# Effect of Honeycomb β-TCP Geometrical Structure on Bone Tissue Regeneration in Skull Defect

**DOI:** 10.3390/ma13214761

**Published:** 2020-10-25

**Authors:** Toshiyuki Watanabe, Kiyofumi Takabatake, Hidetsugu Tsujigiwa, Satoko Watanabe, Ryoko Nakagiri, Keisuke Nakano, Hitoshi Nagatsuka, Yoshihiro Kimata

**Affiliations:** 1Department of Plastic and Reconstructive Surgery, Graduate School of Medicine, Dentistry and Pharmaceutical Sciences, Okayama University, Okayama 7008525, Japan; watanabetoshiii@gmail.com (T.W.); satokot.twatanabe@gmail.com (S.W.); nakagiriryoko@gmail.com (R.N.); ykimata@cc.okayama-u.ac.jp (Y.K.); 2Department of Oral Pathology and Medicine, Graduate School of Medicine, Dentistry and Pharmaceutical Sciences, Okayama University, Okayama 7008525, Japan; keisuke1@okayama-u.ac.jp (K.N.); jin@okayama-u.ac.jp (H.N.); 3Department of Life Science, Faculty of Science, Okayama University Science, Okayama 7000005, Japan; tsuji@dls.ous.ac.jp

**Keywords:** honeycomb β-TCP, bone tissue regeneration, bone microenvironment, Vertical and Horizontal holes, geometrical structure

## Abstract

The effect of the geometric structure of artificial biomaterials on skull regeneration remains unclear. In a previous study, we succeeded in developing honeycomb β-tricalcium phosphate (β-TCP), which has through-and-through holes and is able to provide the optimum bone microenvironment for bone tissue regeneration. We demonstrated that β-TCP with 300-μm hole diameters induced vigorous bone formation. In the present study, we investigated how differences in hole directions of honeycomb β-TCP (horizontal or vertical holes) influence bone tissue regeneration in skull defects. Honeycomb β-TCP with vertical and horizontal holes was loaded with BMP-2 using Matrigel and Collagen gel as carriers, and transplanted into skull bone defect model rats. The results showed that in each four groups (Collagen alone group, Matrigel alone group, Collagen + BMP group and Matrigel + BMP-2), vigorous bone formation was observed on the vertical β-TCP compared with horizontal β-TCP. The osteogenic area was larger in the Matrigel groups (with and without BMP-2) than in the Collagen group (with and without BMP-2) in both vertical β-TCP and horizontal β-TCP. However, when BMP-2 was added, the bone formation area was not significantly different between the Collagen group and the Matrigel group in the vertical β-TCP. Histological finding showed that, in vertical honeycomb β-TCP, new bone formation extended to the upper part of the holes and was observed from the dura side to the periosteum side as added to the inner walls of the holes. Therefore, we can control efficient bone formation by creating a bone microenvironment provided by vertical honeycomb β-TCP. Vertical honeycomb β-TCP has the potential to be an excellent biomaterial for bone tissue regeneration in skull defects and is expected to have clinical applications.

## 1. Introduction

The ability of the calvaria to regeneration is low, and when a bone defect occurs due to trauma or surgical craniotomy and craniectomy, it is difficult for the skull defect to completely heal even in young people. Cranial bone reconstruction requires strength and biocompatibility, thus autologous bone transplantation is clinically the most common method. However, autologous bone transplantation is highly invasive and burdens the patient [1,2]. Additionally, artificial bone that does not self-organize such as hydroxyapatite has recently become used for skull defects. However, these non-resorbable artificial bones block blood flow and the interaction between the artificial bone and adjacent tissues, causing fragility and distortion of surrounding tissues and susceptibility to infection [3,4,5]. In recent years, the usefulness of absorbable ceramics with high biocompatibility such as tricalcium phosphate (TCP) has been shown to be effective for bone tissue reconstruction due to high biocompatibility and high bone tissue induction. TCP is already widely clinically applied [5,6,7,8,9,10].

Artificial biomaterials used for inducing bone formation should possess biocompatibility and bioabsorbability, and extracellular microenvironments created by artificial biomaterials are also considered to play an important role for biocompatibility and bioabsorbability [11]. In particular, it has been reported that the geometrical structure of artificial biomaterials exerts a great influence on the process of hard tissue formation [12,13], and porosity is particularly important for cell proliferation and differentiation and vascular invasion [14]. Generally, when the pores of artificial biomaterials are too large, the stiffness of the porous body becomes weak, and when the pores are too small, the stiffness increases but cells and blood vessels have difficulty entering the pores [12,13,14,15,16]. At present, artificial biomaterials with high porosity have been developed and have already been clinically applied. However, most of their pores are blind ends and have few through-holes. Therefore, these pore structures impair vascular permeability and cell invasion, limiting bone tissue replacement [12,17]. As described above, there are currently no reports of a biomaterial that meets the conditions of cell invasiveness, cell proliferation, cell differentiation, and strength for skull bone reconstruction.

We have focused on the importance of the extracellular microenvironment in the process of hard tissue formation and have been developing new biomaterials with a geometric structure that efficiently induces hard tissue formation. We have confirmed that honeycomb β-TCP, in which linear through-holes with pore diameters of 300 μm are arranged in a honeycomb shape, has very high osteoinductive ability [18]. We also reported that this honeycomb β-TCP, which can reproduce the microenvironment, can control cell differentiation by changing the geometrical structure and enable differentiation of chondrocytes and osteoblasts [19]. Moreover, when we used a rat zygomatic defect model and implanted honeycomb β-TCP into the cheekbone defect, we obtained good bone fusion with the zygomatic stump bone. We found that it was important that the through-holes of honeycomb β-TCP were continuous with the bone marrow cavity for bone reconstruction of zygomatic defects in the facial region [20].

Since the bone marrow cavity is located in the diploic layer in the skull, it is considered that the through-holes of the artificial biomaterial need to be placed continuously with the bone marrow cavity in order to induce efficient bone formation in a skull defect. However, it has been reported that vigorous bone formation occurs even when the through-holes contact the periosteum directly, and there has been no study on differences in skull bone formation depending on the installation direction of the through-holes [21].

Therefore, in this study, we used a rat skull defect model and examined the installation direction of the through-holes to see if it is advantageous for bone formation. Honeycomb β-TCP with linear through-holes was implanted into skull bone defects, and two types of honeycomb β-TCP were prepared: horizontal honeycomb β-TCP with straight through-holes penetrating the interstitial layer of the skull, and vertical honeycomb β-TCP with straight through-holes penetrating the dura and periosteum. Then, the bone-forming ability was compared histologically by transplantation into skull defect model rats.

## 2. Materials and Methods

### 2.1. Preparation of TCP Containing BMP-2

Honeycomb β-TCP was pressed in a rectangular parallelepiped mold with a width of 3.95 mm, a height of 3.95 mm, and a depth of 1 mm, and it contained through-holes with diameters of about 300 μm (Figure 1A). The detailed manufacturing method of β-TCP has been described previously [17]. Each honeycomb β-TCP was sterilized by autoclave and loaded with bone morphogenetic protein-2 (BMP-2), which was diluted to an amount of 1000 ng in Matrigel^®^ (concentration of 80 μg/mL) (BD Bioscience, Bedford, MA, USA) or in Atelocollagen in Eagle’s MEM (Koken, Tokyo, Japan; referred to hereafter as Collagen gel). In the control group, we prepared β-TCP and Matrigel^®^ or Collagen gel without BMP-2.

### 2.2. Animals and Implantation Procedure

A total of 18 four-week-old healthy male Wister rats were used in these experiments. All experiments were performed in accordance with the Policy on the Care and Use of Laboratory Animals, Okayama University, and approved by the Animal Care and Use Committee, Okayama University (OKU-2017019). All surgical procedures were performed under general anesthesia in a pain-free state.

To investigate the honeycomb β-TCP osteoconductivity, the animals were randomly divided into four groups: two different carrier (Matrigel^®^ or Collagen gel) × BMP-2 or without, for a total of four groups. For each group, four Wister rats were used, and one vertical β-TCP and one horizontal β-TCP were implanted into each rat. Additionally, two Wister rats were used for long-term observation, and one vertical β-TCP with Collagen gel + BMP-2 and one horizontal β-TCP with Collagen + BMP-2 were implanted into each rat.

Wistar male rats were anesthetized intraperitoneally with ketamine hydrochloride (75 mg/kg body weight) and medetomidine hydrochloride (0.5 mg/kg body weight), and with atipamezole hydrochloride (1 mg/kg body weight), which was injected subcutaneously when awakening. The region of the head was shaved and cleaned with 70% alcohol and iodine, and then the scalp was cut by 10 mm to expose the skull. The skull was cut using a diamond bar, and two 5 × 5 mm skull tissue defect parts were created. Each sample was implanted carefully in the skull bone defect and sutured (Figure 1B). The animals were killed with an overdose of isoflurane at 4 weeks or 6 months after implantation. For histological observations, implanted TCPs were fixed by perfusion with 4% paraformaldehyde.

### 2.3. Histological Procedure

The specimens were decalcified using 10% ethylenediaminetetraacetic acid for 3 weeks. They were embedded in paraffin and sectioned into 5-μm thicknesses. Sections were chemically stained with hematoxylin and eosin (HE) and observed histologically.

### 2.4. Micro CT

The head specimens after fixation were taken with micro CT (Hitachi Aloka Latheta LCT200, Tokyo, Japan), and the resulting DICOM data was reconstructed three-dimensionally by using the workstation and software (AZE VirtualPlace Lexus64, Tokyo, Japan).

### 2.5. Bone Tissue Formation Evaluation by Area Measurement

To quantify the bone tissue formation area, bone tissue formation was measured in all holes of honeycomb β-TCP in Hematoxylin-Eosin (HE)-stained specimens (200× magnification) using Image J software (NIH, Bethesda, MD, USA). The obtained average value was compared to vertical TCP and horizontal TCP in each collagen (without or with BMP-2) or Matrigel group (without or with BMP-2) (n = 4).

### 2.6. Statistical Analysis

Statistical analysis was performed using one-way analysis of variance and Fisher’s exact tests. A *p* value <0.05 was considered statistically significant. All calculations were performed using PASW Statistics 18 (SPSS Inc., Chicago, IL, USA).

## 3. Results

### 3.1. Bone Tissue Formation in Honeycomb β-TCP Holes

In the Matrigel-loaded (without BMP-2) horizontal honeycomb β-TCP, inflammatory cell infiltration was poor in all TCP holes, and vascular invasion was observed. Bone tissue formation was found only in the lower layer adjacent to the dura, and no bone tissue was found in the periosteal side holes (Figure 2A–C).

In the Matrigel-loaded (without BMP-2) vertical honeycomb β-TCP, bone formation was observed up to the upper part of the TCP holes (Figure 2D–F).

In the Matrigel-loaded (with BMP-2) horizontal honeycomb β-TCP, new bone formation extended to the upper part of the holes and was observed from the dura side to the periosteum side as added to the inner walls of the holes; however, the bone formation ability was weak compared to the vertical honeycomb β-TCP group (Figure 3A–C).

In the Matrigel loaded (with BMP-2) vertical honeycomb β-TCP, vigorous new bone formation was observed up to the upper part along the TCP holes, and almost all vertical holes were filled with vigorous new bone tissue. In addition, the formation of bone marrow-like tissue and vascular cavity were observed in the area surrounded by the bone tissue (Figure 3D–F).

In the bone defect group, bone tissue formation was not observed, and the bone defect area did not change almost even at 4 weeks after implantation (Figure 4A). In Collagen gel-loaded (without BMP-2), there were some holes which did not induce bone formation in both vertical and horizontal TCP (Figure 4B). In Collagen gel-loaded (with BMP-2), both vertical and horizontal TCP were connected with existing bone by new bone formation (Figure 4C).

In Collagen gel-loaded (without BMP-2) horizontal honeycomb β-TCP, inflammatory cell infiltration was poor in all of the TCP pores, and vascular invasion was observed. Only a small amount of bone tissue was found near the dura, and formation of bone tissue was not found in the hole near the periosteal side (Figure 5A–C).

In Collagen gel-loaded (without BMP-2) vertical honeycomb β-TCP, a large number of cells invaded in the TCP holes, and formation of bone tissue was observed in some through-holes from the dura side to the periosteum side (Figure 5D–F).

In the case of the Collagen gel-loaded (with BMP-2) horizontal honeycomb β-TCP, bone tissue formation was observed from the dura side to the periosteum side as if it was added to the inner wall of the hole; however, new bone formation in the horizontal through-holes was rougher than in the vertical through-holes, and there was no bone marrow-like tissue in the horizontal holes (Figure 6A–C).

In Collagen gel-loaded (with BMP-2) vertical honeycomb β-TCP, all holes were filled with vigorous new bone, and the formation of bone marrow-like tissue and vascular cavity was observed in the area surrounded by the new bone tissue (Figure 6D–F).

Long-term observation was performed on a sample obtained by impregnating TCP having vertical holes, which showed strong bone formation for 4 weeks, with collagen gel + BMP-2. Vertical honeycomb TCP was embedded in a rat skull defect, and 6 months later, it was removed, and bone tissue formation in the honeycomb β-TCP hole and absorption of honeycomb β-TCP were observed. Honeycombβ-TCP was partially absorbed, and replacement with bone tissue was observed (Figure 7).

### 3.2. Quantitative Examination of Bone Formation

To investigate the correlation bone tissue formation and structure of honeycomb β-TCP, we quantitatively examined the area of bone formation in holes of honeycomb β-TCP (Figure 8).

In each of the four groups (Collagen gel alone, Matrigel alone, Collagen + BMP-2, and Matrigel + BMP-2), the vertical β-TCP had a larger bone formation area than the horizontal β-TCP. In the Collagen alone group, the Matrigel alone group, and the Collagen + BMP group, the vertical β-TCP had more than twice the bone formation area as the horizontal β-TCP. In the Matrigel + BMP-2 group, the bone formation area of the vertical β-TCP was 1.5 times or more that of the horizontal TCP. The osteogenic area was larger in the Matrigel groups (with and without BMP-2) than in the Collagen group (with and without BMP-2) in both vertical β-TCP and horizontal β-TCP. However, when BMP-2 was added, the bone formation area was not significantly different between the Collagen group and the Matrigel group in the vertical β-TCP.

## 4. Discussion

Various artificial biomaterials such as bioactive glass, Hydroxyapatite (HA), and TCP have been used for skull defects. Among these artificial biomaterials, TCP is superior to other materials in terms of its strength and bioabsorbability. Furthermore, biomaterials other than TCP need the presence of growth factors and stem cells for bone formation, whereas honeycomb TCP do not need their factors for bone formation [22,23,24,25,26,27,28]. In addition, there are many studies focusing on geometrical structures, and it has been reported that it is necessary to provide an appropriate microenvironment such as pore structure for bone tissue formation [4,5,8,9,12,13,14,15,16,18,20,21]. However, although there are some studies that change the pore size of the same material, there is no study that changes the direction of the holes as in this experiment.

In this study, in both horizontal and vertical honeycomb β-TCPs without BMP-2 (both the Matrigel-added group and Collagen gel-added group), continuity between β-TCPs and existing bone was observed, and new bone formation was also observed in many β-TCP holes. We consider the honeycomb β-TCP used here to be a biomaterial with very high osteoconductivity, even though the honeycomb β-TCP did not contain any osteoinductive factor. In our experiments, both horizontal and vertical honeycomb β-TCPs were safe because honeycomb β-TCPs did not induce any infection or inflammation and without clinical complications such as dehiscence and necrosis of the wound site or material exposure. Therefore, we consider the biocompatibility of honeycomb β-TCP to be high.

When BMP-2 was not added to honeycomb β-TCP, vertical honeycomb β-TCP had more bone formation in the holes than did horizontal honeycomb β-TCP, and vertical honeycomb β-TCP showed also new bone formation in almost all holes. However, in both horizontal and vertical honeycomb β-TCPs, bone tissue formation was observed not in the periosteal region, but in the dural region, and vascular invasion was induced from the dural region (Figure 2 and Figure 5).

We previously reported, using a cheekbone reconstruction experiment, that the continuity of existing bone and β-TCP, as well as the continuity of existing bone marrow cavity and bone marrow-like structures formed in TCP holes, are important for osteogenesis [20]. However, in this skull reconstruction experiment, stronger bone formation was observed in the vertical honeycomb β-TCP compared to in the horizontal honeycomb β-TCP, and it was continuous with the bone marrow cavity of the skull and the TCP holes. In addition, in both horizontal and vertical holes, new bone formation was induced from the dura to the periosteum.

Next, BMP-2, which is known to have high osteoinductive activity, was loaded into honeycomb β-TCP, and its bone tissue forming ability was analyzed histologically.

In horizontal honeycomb β-TCP with Matrigel + BMP-2, new bone formation was observed in TCP holes extending from the dura to near periosteum (Figure 3). However, in horizontal honeycomb β-TCP with Collagen gel + BMP-2, vigorous bone formation was observed on the dura side, but no bone formation was observed or there were some holes that did not induce bone formation near the periosteum region (Figure 6).

In contrast, in the vertical honeycomb β-TCP, both the Matrigel + BMP-2 and Collagen gel + BMP-2 groups showed vigorous osteogenesis in almost all holes, and both groups also showed vigorous bone formation from the dura to the periosteum (Figure 3 and Figure 6).

Various materials are currently used as carriers for local delivery of BMP-2 for in vivo bone formation. When BMP-2 alone is impregnated into an artificial biomaterial, BMP-2 rapidly diffuses away from the implantation area, so a delivery carrier is required. Matrigel is widely used as a carrier for growth factors in bone tissue regeneration experiments, and Matrigel has BMP-2 retention and has been used in many osteoinductive animal experiments. Like other experiments, vigorous bone formation was observed in the β-TCP holes in this experimental setup using Matrigel. However, Matrigel is extracted from animals (mouse osteosarcoma), and thus clinical application in humans is actually difficult. Therefore, we also used Collagen gel, which is more applicable to humans.

From the results of our experiments, in the horizontal honeycomb β-TCP, β-TCP loaded into Matrigel had more bone formation than the Collagen gel group, indicating that there was a difference in the retention ability of BMP-2 between Matrigel and Collagen gel. However, in the vertical honeycomb β-TCP, there was no significant difference in bone formation between the two groups, and the geometric structure of the vertical holes fully covered the weakness of BMP-2 retention of the Collagen gel. The combination of a collagen gel that can be used in the human body and β-TCP with vertical holes permits clinical applications in humans.

In this study, we demonstrated that the bone tissue formation started from the dura side both in the horizontal and vertical holes. Bone tissue formation in fractures and wound healing is said to be formed from periosteum [29,30,31]. It has been reported that the osteoblast layer of the periosteum is involved in the healing of fractures, and that mechanical stimulation of the periosteum enhances its osteogenic ability [32,33,34]. However, bone tissue regeneration experiments in a skull defect model have shown that the presence of dura is most important in promoting bone regeneration [35,36,37], and it has been shown that the osteoinductive ability of dura as an accelerator for bone regeneration is superior to that of periosteum.

Greenwald et al. reported that immature dura produce Transforming Growth Factor-β (TGF-β), Fibroblast Growth Factor-2 (FGF-2), and Alkaline Phosphatase (ALP), and that TGF-β1 and FGF-2 act in an autocrine and paracrine manner [38,39]. In this study as well as in our previous study, we showed that the bony tissue was formed from the dura side in both the horizontal and vertical honeycomb β-TCPs, especially in the vertical honeycomb β-TCP, which allows the dura and periosteum to communicate with each other through a straight through-hole, showing more vigorous bone tissue induction than the horizontal honeycomb β-TCP.

## 5. Conclusions

In the present study, new bone formation extended to the upper part of the holes and was observed from the dura side to the periosteum side as added to the inner wall of the holes in vertical honeycomb β-TCP. The combination of a collagen gel that can be used in the human body and β-TCP with vertical holes showed high osteoconductivity and biocompatibility and the vertical honeycomb β-TCP permits clinical applications in humans. Our study indicates that β-TCP with vertical holes is an excellent artificial biomaterial, and β-TCP with vertical holes can serve as a new biological material in the skull region.

## Figures and Tables

**Figure 1 materials-13-04761-f001:**
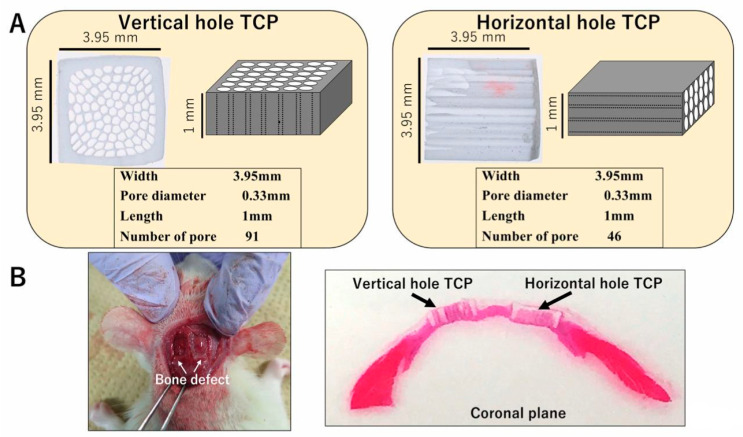
(**A**) Photographs and diagrams of vertical and horizontal β-tricalcium phosphates (β-TCPs). (**B**) β-TCP inserted into the skull bone defect of an experimental animal.

**Figure 2 materials-13-04761-f002:**
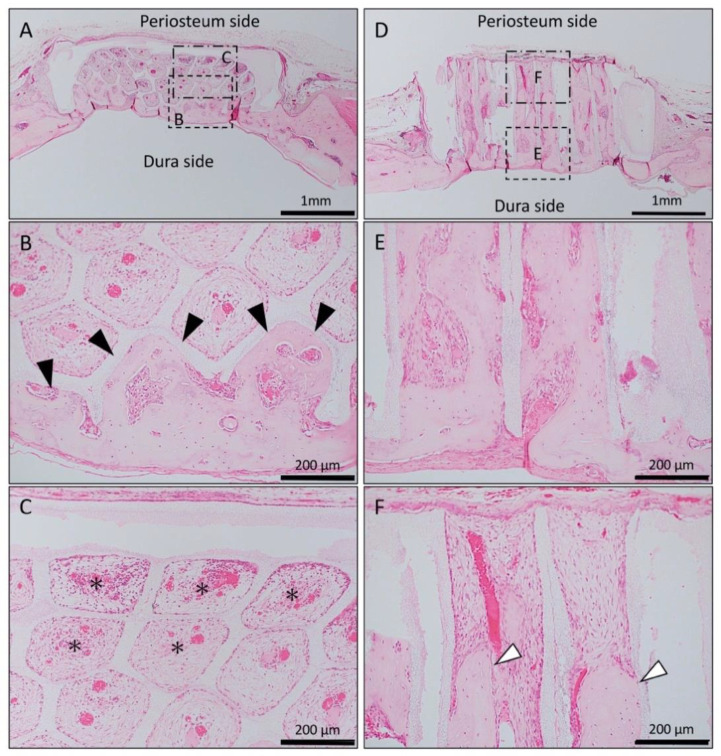
Histological images of β-TCP with Matrigel alone at 4 weeks after implantation. (**A**) Low-magnification image of horizontal β-TCP. No inflammatory granulation tissue was observed around β-TCP. (**B**,**C**) Higher-magnification image of corresponding outlined area in (A). Inflammatory cell infiltration was poor in all TCP holes, and vascular invasion was observed. Bone tissue formation was found only in the lower layer adjacent to the dura (black arrowheads), and no bone tissue was found in the periosteal side holes (asterisks). (**D**) Low-magnification image of vertical β-TCP. No inflammatory granulation tissue was observed around β-TCP. (**E**,**F**) Higher-magnification image of corresponding outlined area in (D). Bone formation was observed up to the upper part of the TCP holes (white arrowheads).

**Figure 3 materials-13-04761-f003:**
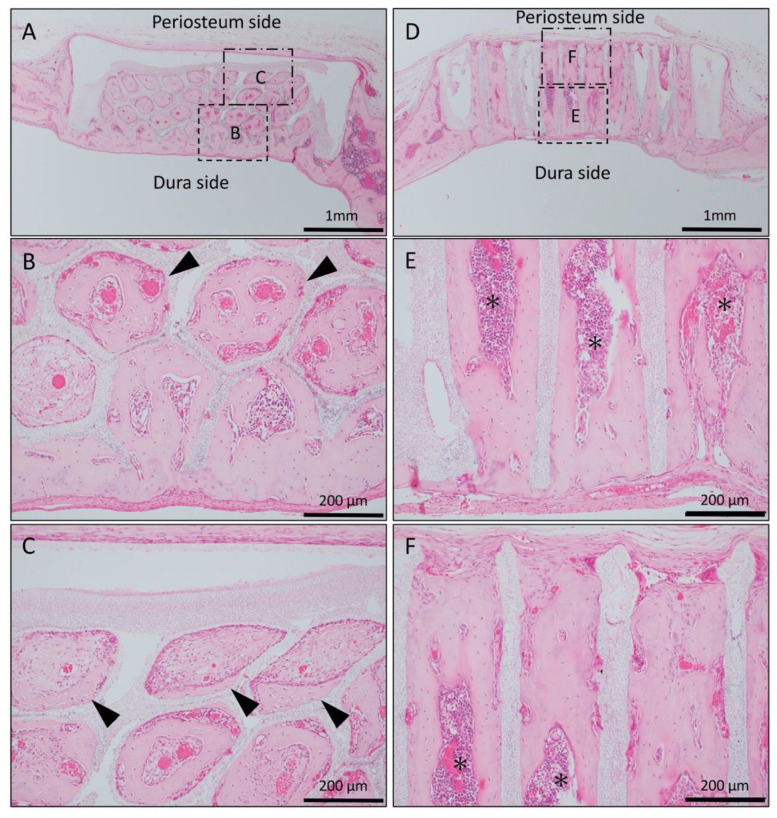
Histological images of β-TCP with Matrigel + bone morphogenetic protein-2(BMP-2) at 4 weeks after implantation. (**A**) Low-magnification image of horizontal β-TCP. No inflammatory granulation tissue was observed around β-TCP. (**B**,**C**) Higher-magnification image of corresponding outlined area in (A). New bone formation extended to the upper part of the holes and was observed from the dura side to the periosteum side as added to the inner wall of the holes (black arrowheads). (**D**) Low-magnification image of vertical β-TCP. (**E**,**F**) Higher-magnification image of corresponding outlined area in (D). Vigorous new bone formation was observed up to the upper part along the TCP holes, and almost all vertical holes were filled with vigorous new bone tissue. In addition, the formation of bone marrow-like tissue (asterisks) and vascular cavity was observed in the area surrounded by bone tissue.

**Figure 4 materials-13-04761-f004:**
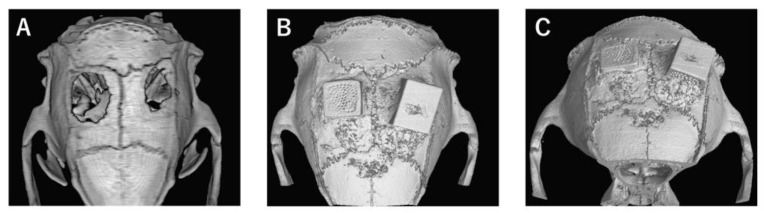
Analysis of bone formation on honeycomb β-TCP in micro CT image in Collagen gel group at 4 weeks after implantation. (**A**) Bone defect group. (**B**) Collagen alone group. (**C**) Collagen+BMP-2 group.

**Figure 5 materials-13-04761-f005:**
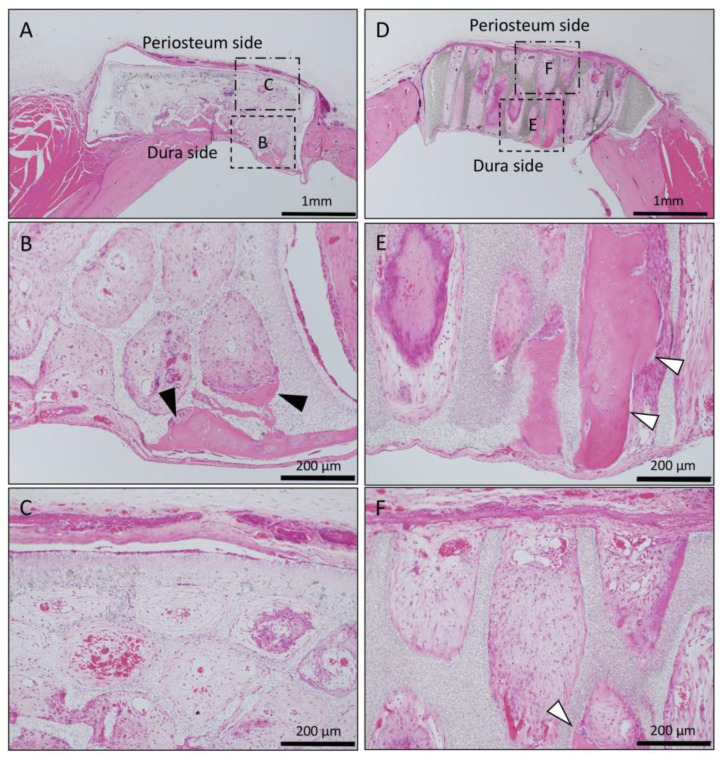
Histological images of β-TCP with Collagen gel alone at 4 weeks after implantation. (**A**) Low-magnification image of horizontal β-TCP. No inflammatory granulation tissue was observed around β-TCP. (**B**,**C**) Higher-magnification image of corresponding outlined area in (A). Inflammatory cell infiltration was poor in all of the TCP holes, and vascular invasion was observed. Only a small amount of bone tissue was found near the dura (black arrowheads), and the formation of bone tissue was not found in the holes near the periosteal side. (**D**) Low-magnification image of vertical β-TCP. No inflammatory granulation tissue was observed around β-TCP. (**E**,**F**) Higher-magnification image of corresponding outlined area in (D). The formation of bone tissue was observed in some through-holes of the dura side (white arrowheads); however, no bone tissue formation was observed in the periosteum side.

**Figure 6 materials-13-04761-f006:**
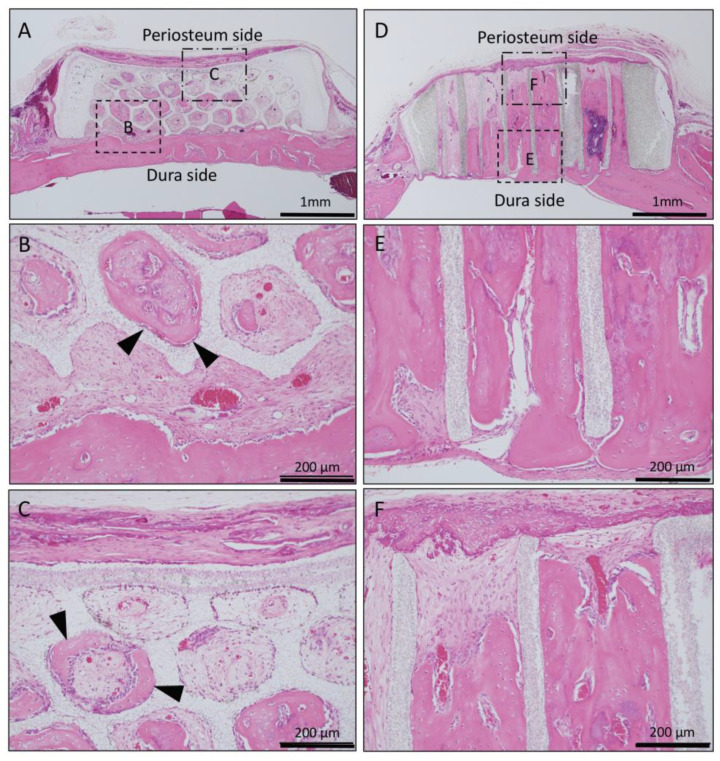
Histological images of β-TCP with Collagen gel + BMP-2 at 4 weeks after implantation. (**A**) Low-magnification image of horizontal β-TCP. No inflammatory granulation tissue was observed around β-TCP. (**B**,**C**) Higher-magnification image of corresponding outlined area in (A). Bone tissue formation was observed from the dura side to the periosteum side as if it was added to the inner wall of the hole (black arrowheads). (**D**) Low-magnification image of vertical β-TCP. No inflammatory granulation tissue was observed around β-TCP. (**E**,**F**) Higher-magnification image of corresponding outlined area in (D). All holes were filled with vigorous new bone, and the formation of bone marrow-like tissue and vascular cavity was observed in the area surrounded by the new bone tissue from the dura side to periosteum side.

**Figure 7 materials-13-04761-f007:**
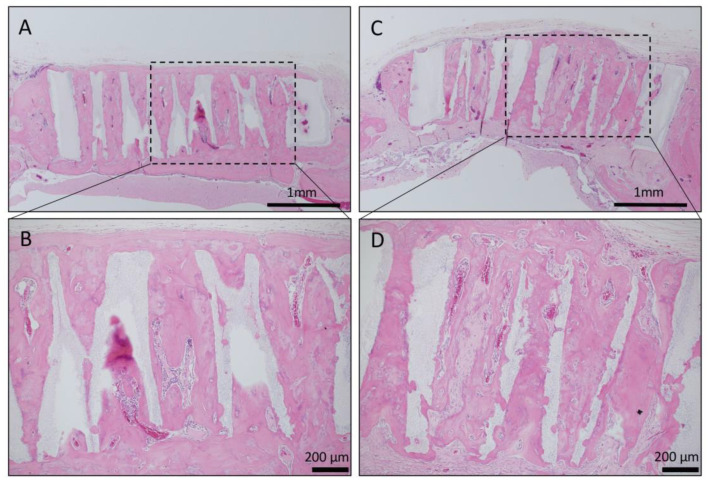
Histological images of β-TCP with Collagen gel + BMP-2 or Collagen gel alone at 6 months after implantation. (**A**) Low-magnification image of vertical β-TCP with collagen alone. Continuity with surrounding existing bone and new bone around β-TCP was observed. (**B**) Higher-magnification image of corresponding outlined area in (A). Vigorous bone formation was observed in the holes of β-TCP; however, the absorption of β-TCP was not observed. (**C**) Low-magnification image of vertical β-TCP with Collagen gel + BMP-2. Continuity with surrounding existing bone and new bone around β-TCP was observed. (**D**) Higher-magnification image of corresponding outlined area in (C). Vigorous bone formation was observed in the holes of β-TCP, and the absorption of β-TCP was also observed.

**Figure 8 materials-13-04761-f008:**
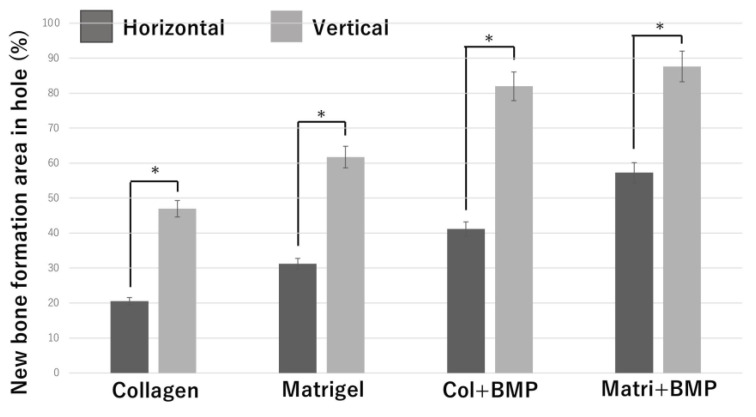
Quantification of the new bone tissue area in the β-TCP holes. *: *p* <0.05.

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
