# Peer review of "Effect of Honeycomb β-TCP Geometrical Structure on Bone Tissue Regeneration in Skull Defect"

_materials, 2020, doi:10.3390/ma13214761_

Round 1
Reviewer 1 Report
- What is the rationale for selecting BMP-2 & incorporating it into beta TCP?
- The study lacks a literature review on the materials used for skull/calvarial defect regeneration. Not just hydroxyapatite, but other biodegradable alloplasts like bioactive glass, calcium sulphate etc have also been used for skull defects. Hence a thorough literature review, with the limitations, if any must be included.
- What is the reason for selecting beta TCP over other bone grafts? Is there any report which indicates that beta TCP has superior performance in skull defect regeneration as compared to other biomaterials?
- Or irrespective of the material, is it the geometric structure that is responsible for inducing bone formation? If so, have you compared beta TCP (horizontal & vertical honeycomb) with other biodegradable alloplasts?
- In Fig. 1B, the cranial defect in the rat model doesn’t seem to be of a square type, rather it looks rectangular. Could you justify the same with higher magnification images?
- Also include pictures showing defects with & without material placement.
- In the methods section, there is no clarity on what are the study groups, how many animals were allotted to each group, how were the materials implanted. For better clarity, the said details can be included in a tabular format.
- The material placement in the defect is also not clear. I presume that a bilateral defect has been created. So are the vertical & horizontal geometric types (for each material) being compared in the same animal? Include these details for better clarity in the experimentation part.
- Why wasn’t a radiographic analysis with bone volumetric analysis performed? A micro-CT analysis would have helped to confirm bone formation along with bone volumetric analysis.
Author Response
1. What is the rationale for selecting BMP-2 & incorporating it into beta TCP?
Response to this comment:
BMP-2 plays an important role in regulating the differentiation of undifferentiated mesenchymal cells into osteoblasts, and has been thought to play a central role in bone regeneration. BMP-2 is widely used as a bone formation promoting factor in in vitro and in vivo bone regeneration studies. Therefore, we selected BMP-2 in this experiment and used with beta TCP.
2. The study lacks a literature review on the materials used for skull/calvarial defect regeneration. Not just hydroxyapatite, but other biodegradable alloplasts like bioactive glass, calcium sulphate etc have also been used for skull defects. Hence a thorough literature review, with the limitations, if any must be included.
Response to this comment:
We have added to the text in Discussion. Line 298-302.
3. What is the reason for selecting beta TCP over other bone grafts? Is there any report which indicates that beta TCP has superior performance in skull defect regeneration as compared to other biomaterials?
Response to this comment:
You have raised an important point. However, in this study, although it is the same material, we have investigated that the bone forming ability differs depending on its geometric structure, so we have not compared with other artificial biomaterials this time.
4. Or irrespective of the material, is it the geometric structure that is responsible for inducing bone formation? If so, have you compared beta TCP (horizontal & vertical honeycomb) with other biodegradable alloplasts?
Response to this comment:
The advantage of honeycomb TCP is that linear through holes arrange with polarity, and this geometric structure has not yet been reproduced in bioactive glass such as alloplasts. Therefore, I did not compare honeycomb TCP with other biodegradable alloplasts.
5. In Fig. 1B, the cranial defect in the rat model doesn’t seem to be of a square type, rather it looks rectangular. Could you justify the same with higher magnification images?
Response to this comment:
We believe that the bone defect size was 5 mm x 5 mm.
6. Also include pictures showing defects with & without material placement.
Response to this comment:
We have added to micro-CT analysis data in Fig. 4.
7. In the methods section, there is no clarity on what are the study groups, how many animals were allotted to each group, how were the materials implanted. For better clarity, the said details can be included in a tabular format.
Response to this comment:
We have mentioned the methods of animals sections in Materials and Methods. (Line115-120)
8. The material placement in the defect is also not clear. I presume that a bilateral defect has been created. So are the vertical & horizontal geometric types (for each material) being compared in the same animal? Include these details for better clarity in the experimentation part.
Response to this comment:
We created a bilateral defect in one same rat and implanted one Vertical and one Horizontal TCP into same rat. We have mentioned the methods in detail in Materials and Methods. (Line115-120)
9. Why wasn’t a radiographic analysis with bone volumetric analysis performed? A micro-CT analysis would have helped to confirm bone formation along with bone volumetric analysis.
Response to this comment:
We have added to micro-CT analysis data in Fig. 4.
Reviewer 2 Report
The manuscript entitled “Effect of Honeycomb β-TCP Geometrical Structure on Bone Tissue Regeneration in Skull Defect” by Watanabe et al. is an interesting study. However, I have some suggestions for the authors to improve their study.
My major concern is the use of only histological analysis to study the effect of Honeycomb β-TCP Geometrical Structure on Bone Tissue Regeneration. It would be important to add other techniques or add immunohistochemical analysis.
Line 50-51: “Artificial biomaterials used for inducing bone formation should possess biocompatibility and bioabsorbability, and extracellular microenvironments are also considered to play an important role [11].” This sentence is not clear.
Lines 57-60: References are missing.
Materials and methods are writing using a different font.
Figure 1 is not clear. Where is the honeycomb structure? Why the authors reported 91 pores for the vertical hole TCP and 46 for the horizontal hole TCP?
Line 92: What is the concentration of BMP-2?
Line 111: the animal were killed with an overdose. Could the authors specify better this point?
Lines 116 and line 119: how many sections were analyzed for each sample?
Lines 120-121: could the authors clarify the number of rats used for each group?
In the result section, the authors should follow always the same order describing the different groups. The authors should describe first horizontal honeycomb β-TCP and then the vertical honeycomb β-TCP. Using the same order for the description will help the reader to follow the manuscript and the correct sequential order of the panels of the figures will also be respected. For example, lines 145-152: Could the author report the data using the same order of the samples without BMP? In this way, also the order of the figure will be correct (first figure 3 A-C and then, figures 3 D-F).
It is not clear why there are no errors bars in figure 7.
The discussion should be better organized. It is difficult to follow and the authors tend to discuss the same points several times.
In the conclusions there is no mention about the use of collagen or matrigel honeycomb β-TCP as well as the use of BMP-2.
Abbreviations should be defined at first mention (for example TGF-β, FGF-2, ALP, TGF-β1 and FGF-2).
Author Response
The manuscript entitled “Effect of Honeycomb β-TCP Geometrical Structure on Bone Tissue Regeneration in Skull Defect” by Watanabe et al. is an interesting study. However, I have some suggestions for the authors to improve their study.
My major concern is the use of only histological analysis to study the effect of Honeycomb β-TCP Geometrical Structure on Bone Tissue Regeneration. It would be important to add other techniques or add immunohistochemical analysis.
Response to this comment:
We have added to micro-CT analysis data in Fig. 4.
Line 50-51: “Artificial biomaterials used for inducing bone formation should possess biocompatibility and bioabsorbability, and extracellular microenvironments are also considered to play an important role [11].” This sentence is not clear.
Response to this comment:
We have modified the sentence. “Artificial biomaterials used for inducing bone formation should possess biocompatibility and bioabsorbability, and extracellular microenvironments created by artificial biomaterials are also considered to play an important role for biocompatibility and bioabsorbability”
Lines 57-60: References are missing.
Response to this comment:
We have added to a reference. Line 60.
Materials and methods are writing using a different font.
Response to this comment:
We have modified the font of Materials and Methods.
Figure 1 is not clear. Where is the honeycomb structure? Why the authors reported 91 pores for the vertical hole TCP and 46 for the horizontal hole TCP?
Response to this comment:
In the actual photograph (vertical hole TCP in Fig. 1A), the straight through holes arrange in a honeycomb shape. The number of holes was different from the vertical TCP and the horizontal TCP because the cross-sectional area of the skull where the horizontal holes were in contact and the area of the skull defect where the vertical holes were in contact were different. Therefore, we could not create the same number holes in between the vertical TCP and the horizontal TCP.
Line 92: What is the concentration of BMP-2?
Response to this comment:
The concentration of BMP-2 was 80 μg/ml. We have added to the concentration of BMP-2 in Line 93
Line 111: the animal were killed with an overdose. Could the authors specify better this point?
Response to this comment:
We used isoflurane instead of ether, so we have modified the text. The concentration of isoflurane was more than 5% and we confirmed cardiac arrest by palpation.
Lines 116 and line 119: how many sections were analyzed for each sample?
Response to this comment:
We analyzed one section for each sample. We have added to the method in detail in Materials and Methods. Line 115-120.
Lines 120-121: could the authors clarify the number of rats used for each group?
Response to this comment:
We used 4 rats for each group. And in long-term observation, we used 2 rats. We have added to the method in detail in Materials and Methods. Line 115-120.
In the result section, the authors should follow always the same order describing the different groups. The authors should describe first horizontal honeycomb β-TCP and then the vertical honeycomb β-TCP. Using the same order for the description will help the reader to follow the manuscript and the correct sequential order of the panels of the figures will also be respected. For example, lines 145-152: Could the author report the data using the same order of the samples without BMP? In this way, also the order of the figure will be correct (first figure 3 A-C and then, figures 3 D-F).
Response to this comment:
We have correct the order of the comments following the figures.
It is not clear why there are no errors bars in figure 7.
Response to this comment:
We have added to the error bars.
The discussion should be better organized. It is difficult to follow and the authors tend to discuss the same points several times.
Response to this comment:
We agree with you and have incorporated this suggestion throughout our paper. We modified the text in Discussion and have deleted the same points several times.
In the conclusions there is no mention about the use of collagen or matrigel honeycomb β-TCP as well as the use of BMP-2.
Response to this comment:
We have modified and added to the text in Conclusion. Line 364-368.
Abbreviations should be defined at first mention (for example TGF-β, FGF-2, ALP, TGF-β1 and FGF-2).
Response to this comment:
We have defined the abbreviation. Line 356-357.
Reviewer 3 Report
The manuscript is well designed and makes an important contribution to the management of skull bone defects. Among the limits:
- The approval of the Ethics Committee is missing;
- It is an interesting work and the data are clear, even if the number of the sample is small and the follow-up is rather limited (from 4 weeks to 6 months).
- The study was done on healthy mice only. Therefore, it would be advisable to evaluate and compare two different homogeneous groups of mice, healthy and diseased (such as arterial hypertension, diabetes mellitus etc.).
Author Response
The manuscript is well designed and makes an important contribution to the management of skull bone defects. Among the limits:
- The approval of the Ethics Committee is missing;
Response to this comment:
We have added to the Ethics Committee code. Line 113.
- It is an interesting work and the data are clear, even if the number of the sample is small and the follow-up is rather limited (from 4 weeks to 6 months).
Response to this comment:
We agree with your comments. We will consider the number of the samples in the future.
- The study was done on healthy mice only. Therefore, it would be advisable to evaluate and compare two different homogeneous groups of mice, healthy and diseased (such as arterial hypertension, diabetes mellitus etc.).
Response to this comment:
We agree with your comments. We will consider that we use honeycomb TCP for diseased. model in the future.
Round 2
Reviewer 1 Report
Although the manuscript has been revised, it still looks incomplete with a lot of missing information and insufficient analysis.
- Rather than just highlighting horizontal or vertical honeycomb β‐TCP, different group considerations i.e. collagen or matrigel loaded with and without BMP‐2 must be included in the abstract & conclusion part. Without the aforementioned details, the abstract is incomplete.
- The authors have added micro-CT analysis only for the collagen group. Not only that, the graft material is superficially seated (raised above the bony margins), and the shape is not confined to the defect margins. Could you please comment on this?
- In the results section, lines 197-201, results pertaining to the collagen group alone have been included. What about the matrigel group? Even in the absence of representative pictures, atleast the corresponding results could have been included.
- In the results section 3.2 Quantitative examination of bone formation, results that have been discussed from the given graph are very scarce. What about the bone formation % comparison among the 4 groups and in the same group, between horizontal & vertical types? The obtained data haven’t been discussed well, which is a major drawback of the current manuscript.
- There are a lot of similar studies in the literature. Hence to highlight the present study, the authors must interpret the obtained results thoroughly with sufficient discussion.
Author Response
Although the manuscript has been revised, it still looks incomplete with a lot of missing information and insufficient analysis.
- Rather than just highlighting horizontal or vertical honeycomb β‐TCP, different group considerations i.e. collagen or matrigel loaded with and without BMP‐2 must be included in the abstract & conclusion part. Without the aforementioned details, the abstract is incomplete.→We have modified the abstract.
- The authors have added micro-CT analysis only for the collagen group. Not only that, the graft material is superficially seated (raised above the bony margins), and the shape is not confined to the defect margins. Could you please comment on this?→In the micro CT images of Fig. 4, bone formation has already been observed in both bone defect group and TCP implantation groups because 4 weeks have passed after the transplantation. Thus, it was observed that the margin of the bone defect was irregular and TCPs were on the bone. We have mention the state of TCPs when TCPs were transplanted to the skull in Fig. 1.
- In the results section, lines 197-201, results pertaining to the collagen group alone have been included. What about the matrigel group? Even in the absence of representative pictures, atleast the corresponding results could have been included.→We have already mentioned the results of Matrigel groups in Fig. 2 and Fig. 3.
- In the results section 3.2 Quantitative examination of bone formation, results that have been discussed from the given graph are very scarce. What about the bone formation % comparison among the 4 groups and in the same group, between horizontal & vertical types? The obtained data haven’t been discussed well, which is a major drawback of the current manuscript.→We have modified the results section 3.2.
- There are a lot of similar studies in the literature. Hence to highlight the present study, the authors must interpret the obtained results thoroughly with sufficient discussion.→We have modified and added to the text in Discussion.
Reviewer 2 Report
A better manuscript after the revision.
The authors added microCT analysis only for the collagen groups. Could the authors add also the images related to matrigel with and without BMP? Could the authors also quantify bone parameters (for example trabecular thickness, bone surface area, bone volume/tissue volume, bone mineral density etc)?
The authors did not mention that they used collagen or matrigel loaded ( with or without BMP‐2) horizontal or vertical honeycomb β‐TCP in the abstract. Also the conclusions of the abstract should be improved adding the differences between collagen and matrigel.
Author Response
A better manuscript after the revision.
The authors added microCT analysis only for the collagen groups. Could the authors add also the images related to matrigel with and without BMP? Could the authors also quantify bone parameters (for example trabecular thickness, bone surface area, bone volume/tissue volume, bone mineral density etc)?
→In this experiment, Collagen gel with TCP that can be applied to the human body was mainly investigated, and Matrigel played a role of positive control or preliminary experiments in this manuscript. Therefore, in this manuscript, Matrigel groups were compared with Collagen gel groups only histologically, and we did not include micro CT images.
Quantify bone parameters: You have raised an important point. We would like to write bone parameters next manuscript.
The authors did not mention that they used collagen or matrigel loaded ( with or without BMP‐2) horizontal or vertical honeycomb β‐TCP in the abstract. Also the conclusions of the abstract should be improved adding the differences between collagen and matrigel.
→We have modified abstract.